# The Effect of the Water Drinking Test on Ocular Parameters and Choroidal Thickness in Glaucoma Suspects

**DOI:** 10.3390/medicina59020381

**Published:** 2023-02-16

**Authors:** Stylianos A Kandarakis, Andreas Katsimpris, Persefoni Kourti, Filippos Psinakis, Efthymios Karmiris, Evangelia Papakonstantinou, Konstantinos Andreanos, Petros Petrou, Ilias Georgalas

**Affiliations:** Department of Ophthalmology, National and Kapodistrian University of Athens, 1st University Eye Clinic, G. Gennimatas General Hospital, 11527 Athens, Greece

**Keywords:** water drinking test, choroidal thickness, glaucoma suspect, optical coherence tomography, intraocular pressure

## Abstract

*Background and objectives*: We aimed to evaluate the effects of the water drinking test (WDT) on several systemic and ocular parameters, including choroidal thickness, which was assessed through optical coherence tomography angiography (OCTA), in glaucoma suspects. *Materials and Methods*: A total of 40 eyes from 20 glaucoma suspects without any systemic or ocular diseases were included in this prospective observational study. All the participants undertook the WDT, which required the drinking of 1 L of table water in 5 min. The outcome measures included IOP, systolic and diastolic blood pressure (SBP and DBP), mean arterial pressure (MAP), mean ocular perfusion pressure (MOPP), ocular pulse amplitude (OPA), and subfoveal and peripapillary choroidal thickness, which were assessed at baseline and at four 15 min intervals after the WDT. Generalized least squares models and mixed model analyses that take into account repeated measurements were used to assess the changes over time of these parameters. *Results*: All the ocular and systemic parameters showed statistically significant changes at all time points compared to baseline apart from choroidal thickness. The peak changes were an IOP of 20.1 mmHg versus 17.3 mmHg at 45 min, an SBP of 137.6 mmHg versus 125 mmHg at 30 min, a DBP of 95.9 mmHg versus 85.7 mmHg at 15 min, and an MOP of 53.51 mmHg versus 48.89 mmHg at 15 min. *Conclusions*: Despite elevations in IOP and significant changes in all the assessed systemic parameters, the WDT was not associated with changes in choroidal thickness in glaucoma suspects.

## 1. Introduction

The water drinking test (WDT)’s application has been subject to debate in regard to its diagnostic and clinical utility over the last 80 years.

Initially, in the 1970s, it failed to meet the criteria of a diagnostic procedure for glaucomatous optic neuropathy because of its low sensitivity and specificity [1,2]. 

Based on recent evidence, it has been gaining popularity as a provocative procedure with high reproducibility for the detection of peak IOP. Several publications have suggested that the peak IOP revealed by the water drinking test is strongly correlated and is in agreement with the IOP peaks that occur during the day [3,4,5,6]. Furthermore, as evidenced by the Collaborative Glaucoma Treatment Study, the WDT is also a factor that predicts long-term glaucomatous progression [7]. A recent longitudinal study confirmed that the IOP peaks that occurred after the WDT had predictive value for future visual field progression in a treated POAG population [8]. 

Considering the recent bibliography, there are different plausible mechanisms which are responsible for the acute IOP increase, one of which being the choroidal engorgement following the water load.

A study by De Moraes et al. [9] related the spikes of IOP to the acute expansion of the choroidal volume in glaucoma patients, and these findings were validated by a more recent study [10]. Furthermore, a study prospectively investigating the changes of the choroidal circulation and structure concluded that the increase in its thickness after the WDT could be attributed, specifically, to the dilation of the lumen of the choroidal vessels [11]. Until today, no studies have investigated whether these changes may be verified in glaucoma suspects.

Therefore, the purpose of our study was to investigate whether similar changes took place in glaucoma suspect patients following the WDT. In addition, we aimed to correlate these changes with variations in choroidal thickness using new imaging techniques.

The advent of the SS-OCTA has enabled better documentation of the choroidal layer. With the application of a longer wavelength (1050 nm) and the presence of the automated segmentation software, variations in the choroidal volume can be assessed in fine detail and with better accuracy [12].

This study aimed to investigate, in glaucoma suspects, the response of systemic and ocular parameters following the water drinking test and, in particular, adds information on the extent of the variation in choroidal thickness and its correlation with IOP and the other variables in question.

## 2. Materials and Methods

This prospective study followed the principles of the Declaration of Helsinki and was approved by the institutional ethics committee of the General Hospital of Athens “G. Gennimatas”.

Individuals who met the criteria to participate in our research project were informed about the study procedure and provided written informed consent.

The investigation was conducted on patients that were referred to our tertiary glaucoma unit. Thirty Glaucoma Suspects who met the study criteria were recruited from the Department of Glaucoma of the 1st Ophthalmology Clinic of the National and Kapodistrian University of Athens from January 2018 to June 2018. All patients were Caucasian; eighteen subjects were female, and twelve were male. 

Glaucoma suspects were defined, according to the American Academy of Ophthalmology Guidelines, as individuals with characteristic clinical features in at least one eye, such as elevated intraocular pressure (IOP ≥ 21 mmHg) on diurnal tension curve or an appearance of the optic disc or retinal nerve fibre layer (RNFL) that is suspicious for glaucomatous damage. In conjunction, they presented normal open angles through gonioscopy, normal visual fields through achromatic automated perimetry, and normal or borderline retinal nerve fibre layers through optical coherence tomography (OCT).

Both visual field examination (Humphrey Field Analyzer 30-2 test, full threshold strategy) and peripapillary RNFL thickness were evaluated on the day of the clinical examination. Unreliable visual fields with more than 33% fixation losses, false-negative errors, or more than 15% false-positive errors were excluded. The thickness of the RNFL was documented using a swept-source technology OCT (1050 nm wavelength light source) and, in particular, the 3D wide scan (12 mm × 9 mm) of the DRI OCT Triton series (Topcon Medical Systems).

The measurement of choroidal thickness was performed with the same instrument, (SS-OCT DRI Triton series) which, due to the optimized long-wavelength scanning light (1050 nm), permits better penetration of the deeper layers of the eye.^16^ Accurate high speed choroidal thickness maps can be produced.

In order to eliminate possible factors that could affect ChT, we had strict inclusion criteria avoiding glaucoma suspects who had any systemic or ocular pathologic condition, were on any systemic medication, or had high refractive errors (>3D). Furthermore, we included subjects that were not using any topical or systemic glaucoma treatment. In particular, exclusion criteria included the use of any antiglaucoma treatment; history of systemic hypertension with or without treatment; history of congestive heart disease; presence of opaque refractive media causing inability to obtain reliable OCT scans; corneal pathology causing inability to measure IOP and OPA; and, finally, refractive error of more than ±3 dioptres. All measurements were obtained at the same time of day, during clinic hours from 1 pm to 3 pm, by two experienced masked technicians. Furthermore, patients who had undergone intraocular surgery within the last year were also excluded from the study. If both eyes of a participant met the inclusion criteria, one was randomly chosen; otherwise, only the glaucoma suspect eye was included in the study.

## 3. Procedure

All subjects were asked to perform water and food deprivation for 3 h before the WDT, and the procedure was performed at the same time of day for all participants (between 1 and 2 pm) with the ingestion of 1 litre of table water within 5 min.

Axial length of the eyes and anterior chamber depth were documented once before the provocative test with the use of the IOLMaster (Zeiss, Jena, Germany).

The rest of the ocular and systemic parameters were measured before water ingestion to establish a baseline (0 min) and at 15, 30, 45, and 60 min after water ingestion.

Intraocular pressure was documented using both the Goldmann applanation tonometer (GATT, Haag-Streit, Switzerland) and the DCT (Pascal dynamic contour tonometer, Swiss Microtechnology AG, Port, Switzerland). To avoid corneal and angle deformation, which could lead to false IOP results, Pascal tonometry preceded GATT by 2 min, and only measurements with a quality score of 1 and 2 (Q1/Q2) were accepted.

Both techniques of IOP measurement were carried out at each time point by two experienced examiners following the same order (FP and KA).

Blood pressure (BP) was measured with a brachial sphygmomanometer (MDF 840, MDF instruments) and a single head stethoscope (MDF 740) after the individuals were seated for at least 5 min. Two measurements were taken at an interval of 5 min, and the average value was recorded.

The vascular factors related to glaucomatous neuropathy, such as mean ocular perfusion pressure (MOPP) and ocular pulse amplitude (OPA), were recorded as well. OPA values were given by the Pascal tonometer, and MOPP was calculated with the formula MOPP = 2/3 MAP − IOP (MAP = mean arterial pressure), where mean arterial pressure was calculated with the formula MAP = DBP + [1/3 × (SBP − DBP)] (DBP = diastolic blood pressure, and SBP = systolic blood pressure).

Finally, ChT was measured by two experienced technicians at baseline and at each time point following the WDT (15′, 30′, 45′, and 60′). As final measurement, the mean value of the two measurements was used. Subfoveal and peripapillary choroidal thickness were measured using Topcon automated OCT software. Peripapillary thickness consisted of four zones (superior, inferior, temporal, and nasal). The demarcation of the choroidal borders in each location was completely automated.

## 4. Statistical Analysis

Microsoft Excel was utilized to gather the data, and all the statistical analyses were performed using statistical software R (version 3.5.1, Foundation for Statistical Computing, Vienna, Austria). The baseline descriptive statistics of the study participants were reported as percentage values for the categorical variables and as mean ± standard deviation (minimum–maximum) for the continuous variables. To detect the changes over time in IOP, systolic and diastolic blood pressure, mean ocular perfusion pressure, and mean arterial pressure, generalized least squares models were used (with a compound symmetry correlation structure) that take into account repeated measurements. Since the choroidal thickness of both the eyes of the participants was assessed, a linear mixed model analysis with a random intercept on the individual level was used to assess the relationship between choroidal thickness and the minutes after the WDT. Therefore, by using a mixed model analysis, we adjusted for the relationship between the choroidal thickness of different eyes from the same participants. *p* values less than 0.05 were considered statistically significant, and all the statistical tests were two-sided. Data extraction in Microsoft Excel was performed separately by two authors (SAK and AK), who did not find any discrepancies in their final documents. Moreover, we performed manual data validation by manually inspecting the column cells as well as by utilizing the data validation tools in Excel.

## 5. Results

A total of 20 patients (40 eyes) met the inclusion criteria and were enrolled in our study between 2016 and 2017. The patients were all Caucasian; 14 were female, and 6 were male. The mean age of the individuals considered to be glaucoma suspects was 65.45 years (±8.2). A total of 7 subjects were pseudophakic, and 13 were phakic. The average anterior chamber depth was 3.2 mm (±0.7), and the mean axial length was 23.3 mm (±1.0). The demographics of the patients included in the study are summarized in Table 1. 

Following the water load, dynamic contour tonometry (Pascal) showed a statistically significant rise at all the time points compared to baseline, with peaks of their mean values at 45 min (20.1 mmHg ± 3.4 mmHg) (*p* < 0.05).

The Goldmann applanation tonometry values were higher than those at baseline at all the time points but only reached statistical significance at 30 min (18.4 mmHg ± 4.5 mmHg) and 45 min (18.4 mmHg ± 4.4 mmHg) (*p* < 0.05).

OPA increased from a baseline value of 2.4 mmHg (±1.0) to 2.7 mmHg (±1.1) at 15 min after the WDT and then remained stable. The increase was statistically significant at all the time points. The mean values of IOP both with Goldman applanation tonometry and Pascal tonometry and of OPA are included in Figure 1.

Systolic blood pressure and diastolic blood pressure were significantly increased at all the time points (15′, 30′, 45′ and 60′ after the WDT) compared to baseline (*p* < 0.05). Peak mean systolic pressure (137.6 mmHg ± 10.3) was reached at 30 min, whereas peak mean diastolic pressure (95.9 mmHg ± 10.7) was reached at 15 min after the WDT. (Figure 2).

Mean arterial pressure was increased at a statistically significant level compared to baseline at each time point after the WDT. From an average baseline value of 99.66 ± 9.45 mmHg, MAP was 109.69 ± 8.68 mmHg at 15′, 108.98 ± 8.26 mmHg at 30′, 108.21 ± 10.27 mmHg at 45′, and 106.25 ± 6.29 mmHg at 60′ following the WDT.

Mean ocular perfusion pressure increased from a baseline average value of 48.89 ± 5.73 mmHg to 53.51 ± 5.26 mmHg at 15′ and remained mostly stable until the 45′ time point. Finally, at the 60’ time point, MOPP started decreasing with an average value of 52.73 ± 5.71 mmHg.

We found no statistically significant associations between mean peripapillary and subfoveal choroidal thickness and the time after the water drinking test (Table 2), with the association estimates ranging from a −0.20 μm (95% CI: −1.71 to 1.31) to 0.67 μm (95% CI: −0.66 to 1.99) change in choroidal thickness per 10 min increment after the water drinking test. Scatterplots and regression lines of the above-mentioned associations can be found in Figure 3 and Figure 4.

## 6. Discussion

The WDT is an acute-load provocative test that can indirectly provide information about the outflow facility of the eye [3,13]. It is well documented that, both in normal subjects and in glaucoma patients, it causes a remarkable and transient IOP increase of a variable degree.

Furthermore, recently published studies demonstrate that it can predict, with accuracy and high reproducibility, peak circadian IOP [5]. What is not specified yet is the exact mechanism behind the acute increase in intraocular pressure.

The pattern of the IOP response in our cohort of glaucoma suspects was similar to that observed in previous studies with different selected populations [14] (normal subjects, open-angle glaucoma patients, angle-closure glaucoma patients). In particular, we documented, with both tonometric techniques (GATT/DCT), a gradual rise in mean IOP from baseline to the 45′ time point. Thereafter, IOP decreased but remained higher than that at baseline.

Four possible causes have been suggested to explain this transient intraocular pressure rise.

The first theory suggests that a disturbance in the blood–ocular osmotic pressure gradient leads to the hydration of the vitreous humour followed by increased aqueous humour production. However, a second potential explanation behind the transient increase in IOP is that, following the rapid ingestion of water, there is an alteration in the filtration capacity of the trabecular meshwork.

The last two possible mechanisms attribute the rise in IOP to systemic and local vascular factors. Following the water load, there is an increase in central and peripheral venous pressure that could lead to an increase in episcleral venous pressure and also to the engorgement of the choroidal vasculature.

Although, during the last couple of years, new insights about some of the physiologic mechanisms behind this acute and transient IOP response have become more evident, until now, none of these theories has been definitely established, and it seems possible that more than one mechanism could be responsible [15,16,17].

On the other side, one of the principal functions of the choroid is the modulation of the intraocular pressure via the vasomotor control of the blood flow, possibly through intrinsic choroidal neurons [18,19].

The advent of the SS-OCTA has allowed us to quantify in fine detail choroidal thickness in an objective, operator-independent manner and permitted us to document the specific pattern of variations in choroidal thickness following the WDT.

We showed that, following the WDT, although there was a consistent and statistically significant increase in IOP at most of the time points, both with GATT and with DCT, the choroid did not present a significant increase in its thickness. Particularly, there was a specific pattern of choroidal thickness variation which was not correlated with the increase in IOP.

A recent study by Mansouri et al. [10] in healthy subjects evaluated the changes in choroidal thickness (ChT) following the WDT with the use of the SS-OCT and an automated segmentation software and was in accordance with our results. Although there was a statistically significant increase in the thickness of the choroid, this was not correlated with the rise in IOP. It is important to note that our study verified similar findings in glaucoma suspect patients.

Another study by Arora et al. [20], documenting the variation in ChT in glaucomatous patients at 30’ following the WDT using the enhanced depth imaging (EDI) technique on a spectral domain (SD) OCT, concluded that the IOP rise was not fully explained by the ChT increase. They did not find a significant correlation between the absolute or percentage IOP rise and the absolute or percentage ChT change. This phenomenon was attributed to the coincident rise in both IOP and BP, which, in turn, caused OPP (OPP = BP − IOP) to remain stable, possibly leading to an unmodified ChT. However, the manual delimitation of the choroid in a limited area of 6 mm possibly reduced the accuracy and did not permit a detailed segmental evaluation of ChT in its various sectors.

It is well documented that variations in IOP, BP and OPP, due to various factors, could affect the dynamic nature of choroidal thickness [21,22].

Even though, after the WDT, IOP increased, Arora et al. [20] documented a decrease in ChT in some subjects at the 30 min time point, but they did not analyse this behaviour further.

De Moraes et al. [9] evaluated the modification of ocular parameters following the WDT in 20 glaucomatous patients. They used an ocular ultrasonographic technique (A and B scans) to document ChT and observed that, after the water load, there was a significant increase in OPA and ChT which preceded the IOP rise. This observation led to the hypothesis that the IOP elevation following the WDT could be attributed to the expanded choroidal volume due to hemodynamic forces.

A recent study [23] evaluated the change in IOP and ChT after the WDT in patients with pseudoexfoliation syndrome. Researchers used the EDI technique on an SD-OCT and documented the choroidal change exclusively in the subfoveal area. They concluded that there was no correlation between the IOP rise and the ChT increase and that choroidal expansion could not be the only causative factor of the IOP increase following the WDT in this cohort of patients with pseudoexfoliation.

Our secondary purpose was to investigate, in response to the water load, the variations of the two vascular parameters related to glaucoma (OPP−OPA) and to document their correlation with the modification of ChT.

Previous studies evaluated the modification of systemic and ocular hemodynamic parameters following the WDT. Most of them were held in healthy or glaucomatous subjects. In the present cohort of glaucoma suspects, we observed that the vascular factors studied were also altered.

Systolic blood pressure and diastolic blood pressure were significantly increased at all the time points, and the resulting MAP was also increased. Recent studies [24,25,26] showed that the acute water load has a profound effect on BP due to two mechanisms. The first appears to be mediated through the activation of the autonomic nervous system, and the second is associated with hypervolemia caused by the water ingestion. The literature suggests that autonomic activation is the major contributor with the activation of both the sympathetic and parasympathetic systems.

Moreover, confirming the parasympathetic activation following the WDT, a study [27] on healthy subjects attributed the increase in IOP to the collapse of Schlemm’s canal, which was documented through anterior segment OCT. As the thickness of the trabecular meshwork was unchanged whilst the heart frequency, which reflected parasympathetic activity, was found to increase significantly after the water load, which was in conjunction with the previous knowledge of the presence of the parasympathetic nerves in the regulation of the lumen of Schlemm’s canal, the authors concluded that the transient increase in IOP after the WDT could be attributed to the collapse of the canal, which was mediated through parasympathetic activation.

Another study [28] evaluated the level of IOP and the diameter of Schlemm’s canal following the cold pressor test (CPT), which activates the sympathetic nervous system. They verified that sympathetic activation leads to the dilation of Schlemm’s canal and a reduction in IOP.

MOPP was significantly increased at all the time points after the water load. As MAP increased proportionally more than IOP, OPP was increased as well and remained increased until the 45′ time point. One hour after the water load, it started decreasing.

Since ChT seems to be influenced by OPP [29], the pattern of the modification of the choroid that we observed following the WDT could be related to the particular temporal variations in MAP with respect to IOP.

OPA is believed to represent a surrogate of the uveal blood volume during each cardiac circle [30].

Following the water load, the OPA values increased significantly from baseline and remained increased until the 60′ time point. This was reasonable since the ocular pulse is created by the pulsatile blood flow in the choroid, which, in turn, is increased following the hypervolemia and the increased vascular resistance caused by the autonomic activation provoked by the acute water load.

Furthermore, the increased values of OPA were positively associated at a significant level (r = 0.47, *p* = 0.016) with the increased thickness of the superior peripapillary choroid, but they did not correlate at a statistically significant level with the variation in the other choroidal areas.

Our results regarding the OPA values were aligned with previous published studies [30,31,32] documenting that there is no correlation with systemic hemodynamic parameters as expressed by systolic and diastolic blood pressure.

A limitation of this study was the relatively low number of participants. This was a conscious decision as we adhered to strict criteria in order to minimize the possible parameters that could interfere with the outcome. For this reason, patients under glaucoma or antihypertensive treatment were not included. Other limitations included the sample of the patients since they were all Caucasian and were all identified as glaucoma suspects. Our results should only be interpreted for similar population groups since significant changes might apply to patients with different backgrounds.

The strengths of the study were its prospective design; the use of an objective automatic segmentation programme for the documentation of the choroidal boundaries; and the measurement of choroidal thickness at the same time for all the participants, minimizing the possible circadian variations in choroidal thickness.

In conclusion, in this cohort of glaucoma suspects, both ocular and systemic parameters were modified after the WDT. We did not succeed in observing any correlation between the increase in OPA or IOP (both with GAP and with DCT) and the modification of choroidal thickness as documented with the SS-OCTA.

## Figures and Tables

**Figure 1 medicina-59-00381-f001:**
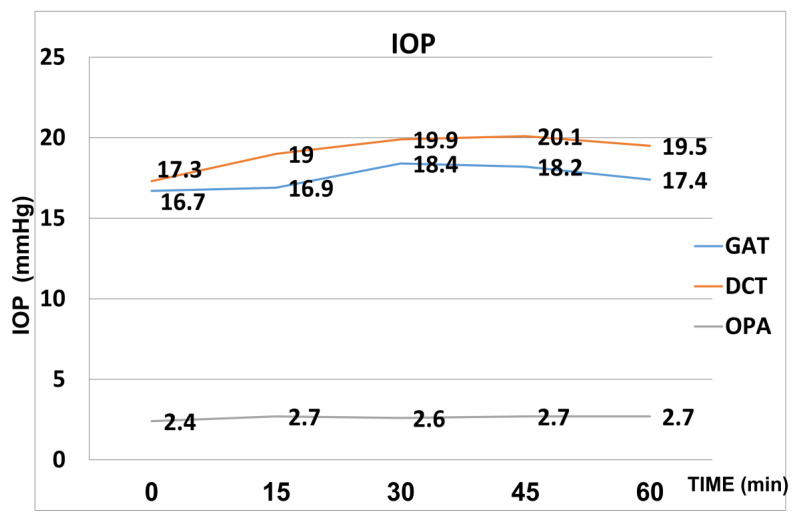
Mean values of IOP (Goldmann and Pascal) and OPA after the WDT.

**Figure 2 medicina-59-00381-f002:**
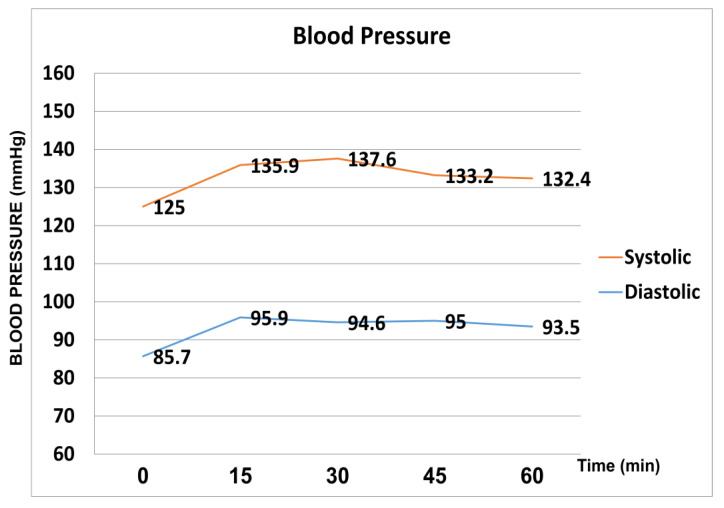
Systolic and diastolic pressure fluctuation during the water drinking test.

**Figure 3 medicina-59-00381-f003:**
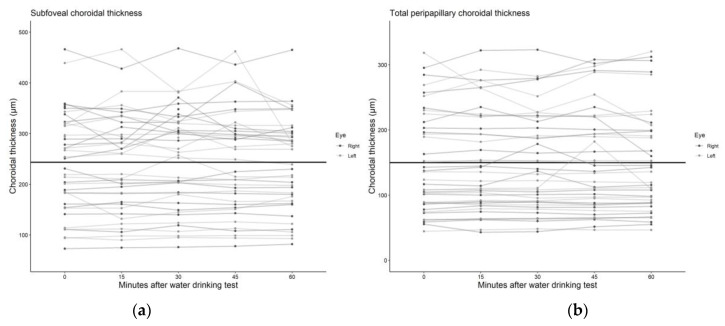
Scatter plot and regression line of the mean change in (**a**) subfoveal and (**b**) total peripapillary choroidal thickness after the water drinking test.

**Figure 4 medicina-59-00381-f004:**
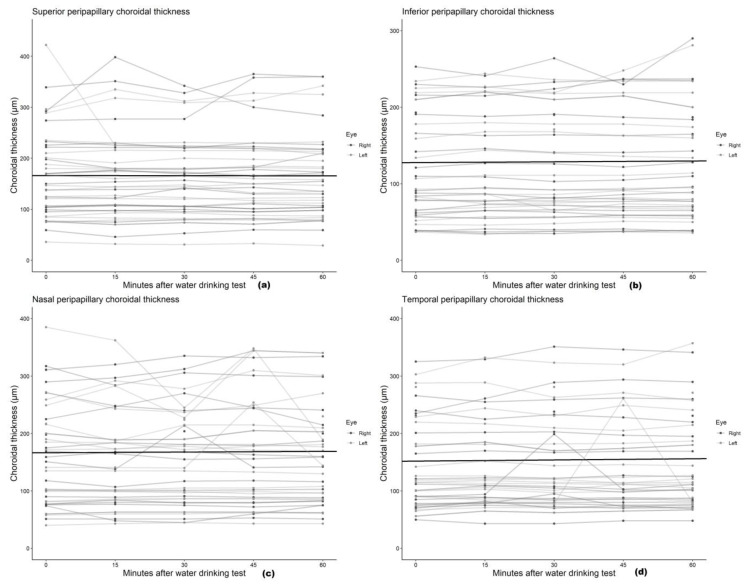
Scatter plot and regression line of the mean change in (**a**) superior, (**b**) inferior, (**c**) nasal, and (**d**) temporal peripapillary choroidal thickness after the water drinking test.

**Table 1 medicina-59-00381-t001:** Baseline characteristics of study population (*n* = 20 patients/40 eyes) ^a^.

Parameters	Values
Age (years)	
Mean ± SD	65.45 ± 8.24
Median (min–max)	68.00 (48–77)
Age group (years), n (%)	
<50	1 (5)
50–59	5 (25)
60–69	5 (25)
>70	9 (45)
Sex, n (%)	
Women	14 (70)
Men	6 (30)
Status	
Phakic	13 (45)
Pseudophakic	7 (35)
ACD	3.2 ± 0.7 mm
AL	23.3 ± 1 mm

^a^ SD represents standard deviation, ACD represents anterior chamber depth, and AL represents axial length.

**Table 2 medicina-59-00381-t002:** Association estimates of changes in mean choroidal thickness (CT) after the water drinking test ^a^.

Variable	Mean Choroidal Thickness (μm)					Regression Coefficient (95% CI)	*p*-Value
	Baseline	15 min	30 min	45 min	60 min		
Subfoveal CT	244	239	247	244	241	−0.10 (−2.24 to 2.04)	0.926
Peripapillary CT	148	141	148	144	148	0.13 (−1.51 to 1.79)	0.873
Superior	165	161	162	160	164	−0.20 (−1.71 to 1.31)	0.793
Inferior	121	119	122	119	124	0.35 (−0.31 to 1.01)	0.305
Nasal	161	156	159	165	160	0.40 (−1.25 to 2.05)	0.637
Temporal	144	142	149	146	148	0.67 (−0.66 to 1.99)	0.325

^a^ Regression coefficients were derived from mixed effect models with random intercepts adjusted for sex and expressed as mean changes in choroidal thickness per 10 min increment after the water drinking test.

## Data Availability

Data available upon request.

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
