# Peer review of "The Effect of the Water Drinking Test on Ocular Parameters and Choroidal Thickness in Glaucoma Suspects"

_medicina, 2023, doi:10.3390/medicina59020381_

Round 1

Reviewer 1 Report

The manuscript is well-designed and well-written. The originality consists of measuring a parameter to verify the correlation between choroidal thickness and WDT in glaucoma patients, which can explain some aspects of the pathophysiology of the disease. 

Regarding a similar topic, all studies discuss possible causes of the increase in IOP following the water load. However, in this work, the cohort study was homogeneous and measured simultaneously, minimizing circadian variation.

Please you would choose between GATT (lines 123 and 125) and GAT, correct GAP (line 355) and introduce DCT

Author Response

We would like to thank the reviewer for his comments and we are very pleased that he believes our work is contributing to current literature. All minor corrections have been addressed to the manuscript.

Reviewer 2 Report

in the statistical analysis chapter reference is made to the use of an excel spreadsheet to collect the data. Please confirm the precautions taken to ensure data reporting and maintain data integrity during statistical analyses. What are the validations made to validate the calculation formulas of the table? specify.

the point clouds of figures 3 and 4 are not readable. These point clouds look extremely scattered. It would be interesting to give the evolution of the parameters studied by patients over time and to make statistical calculations on these evolutions by patients. This could avoid statistical biases, especially with the small number of patients studied. 

Author Response

Comment #1

In the statistical analysis chapter reference is made to the use of an excel spreadsheet to collect the data. Please confirm the precautions taken to ensure data reporting and maintain data integrity during statistical analyses. What are the validations made to validate the calculation formulas of the table? specify.

Answer:

Dear Reviewer, thank you for your valuable comments We agree with your criticism. Data extraction in Microsoft Excel was done separately by two authors (SAK and AK), who did not find any discrepancies in their final documents. Moreover, we performed manual data validation by manually inspecting column cells as well as by utilizing the data validation tools in Excel. We added a relevant comment in the Statistical analysis section of the manuscript.

Comment #2: the point clouds of figures 3 and 4 are not readable. These point clouds look extremely scattered. It would be interesting to give the evolution of the parameters studied by patients over time and to make statistical calculations on these evolutions by patients. This could avoid statistical biases, especially with the small number of patients studied. 

Answer:

We thank the reviewer for this comment. In order to improve the readability of the individual parameter changes through time we revised our Figures by adding lines connecting the values of each individual through time. The revised Figures depicting the individual changes in choroidal thickness through time make it clear, that choroidal thickness values within each patient are quite similar, but there's a lot of difference among individuals and so values of the choroidal thickness are largely determined by which individual the unit belongs to. This is also confirmed by the variance partition coefficient at the individual level (the choroidal thickness variance at the individual level of the statistical model, divided by the total choroidal thickness variance) which ranges from 87.7% to 97.9% in our statistical models. By using mixed model analysis, we adjusted for the repeated choroidal thickness measurements of each individual through the 5 different time points (0, 15, 30, 45 and 60 minutes).and the association estimates reported in Table 2 are derived from these valid models.